# A Survey of Data Representation for Multi-Modality Event Detection and Evolution

Kejing Xiao [1], Zhaopeng Qian [2] and Biao Qin [1,*]

1 School of Information, Renmin University of China, Beijing 100872, China; xiaokejing@ruc.edu.cn
2 School of Artificial Intelligence, Beijing Technology and Business University, Beijing 100048, China; qianzhaopeng@btbu.edu.cn
* Correspondence: qinbiao@ruc.edu.cn

**Featured Application: Specific application include detect hot events and monitor the development of events, potential applications include public opinion regulatory, monitor emergencies and multi-modality information processing.**

**Abstract:** The rapid growth of online data has made it very convenient for people to obtain information. However, it also leads to the problem of "information overload". Therefore, how to detect hot events from the massive amount of information has always been a problem. With the development of multimedia platforms, event detection has gradually developed from traditional single modality detection to multi-modality detection and is receiving increasing attention. The goal of multi-modality event detection is to discover events from a huge amount of online data with different data structures, such as texts, images and videos. These data represent real-world events from different perspectives so that they can provide more information about an event. In addition, event evolution is also a meaningful research direction; it models how events change dynamically over time and has great significance for event analysis. This paper comprehensively reviews the existing research on event detection and evolution. We first give a series of necessary definitions of event detection and evolution. Next, we discuss the techniques of data representation for event detection, including textual, visual, and multi-modality content. Finally, we review event evolution under multi-modality data. Furthermore, we review several public datasets and compare their results. At the end of this paper, we provide a conclusion and discuss future work.

**Keywords:** event detection; event evolution; data representation; multi-modality

## 1. Introduction

With the development of various online platforms, such as news media platforms (e.g., google news), social media websites (e.g., Twitter) and image/video sharing platforms (e.g., Flickr), users can conveniently get information from the internet or share texts, images or videos anytime and anywhere by using their smartphones. However, the extensive development of information platforms has created the problem of "information overload". Information changes rapidly and people witnessing or involved in events find it difficult to find valuable topics amongst the massive information. Therefore, it is a problem for people to find meaningful topics from the massive online data. One solution to this problem is called Topic Detection (TD), which focuses on mining real-world occurrences in unprecedentedly vast online data. Topic detection is originated from the Topic Detection and Tracking (TDT) task and gradually became a hot research topic with the increase of online data [1]. Generally, a topic is defined as something non-trivial happening at a specific time or place [2]. Topic detection techniques can be applied in many realistic scenarios, such as reputation monitoring [3], public opinion monitoring [4] and emergency management [5]. In most cases, "topic" and "event" can be exchanged [6]. So, the two words can be used interchangeably in this paper.

Although there are many studies about event detection on single modality data and have achieved excellent performance, these methods cannot deal with multi-modality data because the multi-modality data are more complex than single-modality data. It contains textual, image or even video data types. Figure 1 gives an example of multi-modality data, in which the left picture shows a political event that composed of textual and image data types, and the right picture shows a sport event that composed of textual and video data types. Compared with the single data form and limited information of single-modality data, multi-modality data can represent real-world events from multiple perspectives and thus provide more comprehensive information of the events. Therefore, many researchers began to conduct research on multi-modality data. However, multi-modality event detection is of great challenge, for that the large amount of multi-modality information has the characteristics of cross platform, multiple modalities, large scale and information redundancy. Different modalities such as text and visual have the "media gap", which make the representation of different modalities in different dimension and cannot be measured directly.

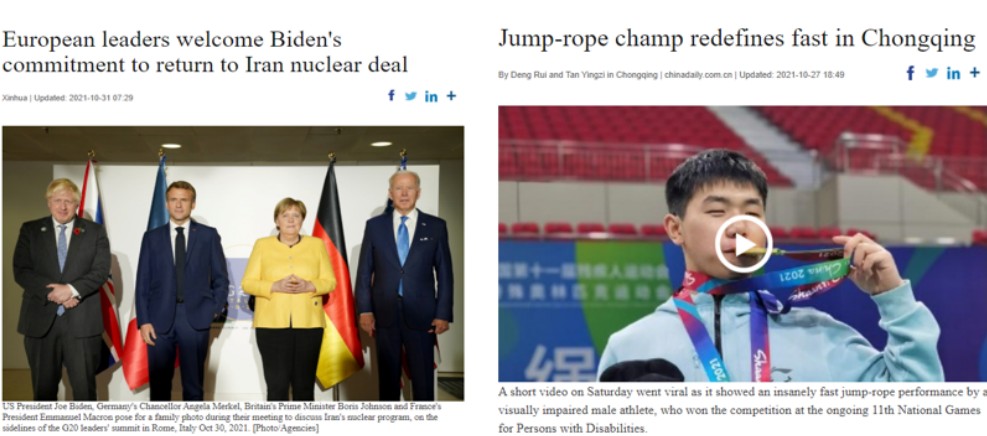

**Figure 1.** Example of multi-modality data.

In recent years, many researchers have surveyed event detection. For example, some works surveyed event detection techniques by the textual data on social media data, which is only single-modality data [7–10]. Goswami A et al. [11] reviewed the event detection methods based on several different online platforms. Tzelepis C et al. [12] surveyed event detection in audio and video. However, they discuss event detection for different modality separately, rather than combining the various modalities. Zeppelzauer M et al. [13] gave various experiments results of event classification based multi-modality data. However, they mainly focused on event classification and event relevance detection on their designed experiments. Zhou H et al. [14] mainly reviewed the research of topic evolution maps based on cross-media data from the perspective of topic modeling. Liu T et al. [15] reviewed the feature learning and event inference techniques. Zhou H et al. [16] surveyed multi-modal social event detection. However, they only introduced multimodal data representation methods, but failed to combine data representation methods with event detection in some reviewed works.

This paper gives a comprehensive analysis of multi-modality event detection and evolution. We start with event representation technologies, in which the single-modality and multi-modality based methods are all surveyed. In each section, we combine the event detection methods with data representation methods. Then, we review event evolution under multi-modality data. Subsequently, several multi-modality event detection datasets are introduced. Based on these datasets, the results comparison between different methods is presented. Comparison results show that multi-modality event detection can achieve better results than single-modality event detection on multi-modality datasets. Finally, we provide a discussion to analyze the development of multi-modality event detection in the future.

In the rest of the review, Section 2 introduces the methodology, Section 3 gives the notion of event detection and evolution. Single-modality data representation and event detection including text based methods and visual based methods are presented in Section 4. Multi-modality data representation and event detection methods are presented in Section 5. Section 6 discusses multi-modality event evolution. The public datasets and results comparison are provided in Section 7. Section 8 gives the conclusion and Section 9 discusses future work.

## 2. Methodology

### 2.1. Retrieval Strategy

This review is organized under the guidelines of Preferred Reporting Items for Systematic Reviews and Meta-analyses (PRISMA). We retrieval the well-known academic databases including IEEE Xplore, Web of Science, Google Scholar and Engineering Village. The keywords used for the retrieval processes include "event detection", "topic detection", "event evolution", "photo event detection", "video event detection" and "multi-modality event detection". The retrieval time range is between 1 January 2000 and 30 June 2021, and all the papers are published in English. In case of omission, some of the literature was searched manually and confirmed by experts.

### 2.2. Selection Strategy

To reduce the possible bias in the selection process, all the authors worked together to decide the selection strategy. Based on the retrieval results, we first excluded papers that have not been cited by researchers as their contribution may be low. Then we excluded papers with insufficient relevance; all the three authors worked together to decide whether a literature is relevant to our study by experience. Finally, 114 papers were included in our review, most of them are journal articles and conference articles published after peer review. There are also two ArXiv articles and their citation times are very high, indicating that they are of high value, so they are also included in our review. All the papers are related to the different representation techniques applied for single-modality event detection, multi-modality event detection and multi-modality event evolution. After selection, the publication distribution of each kind of method is shown in Figure 2.

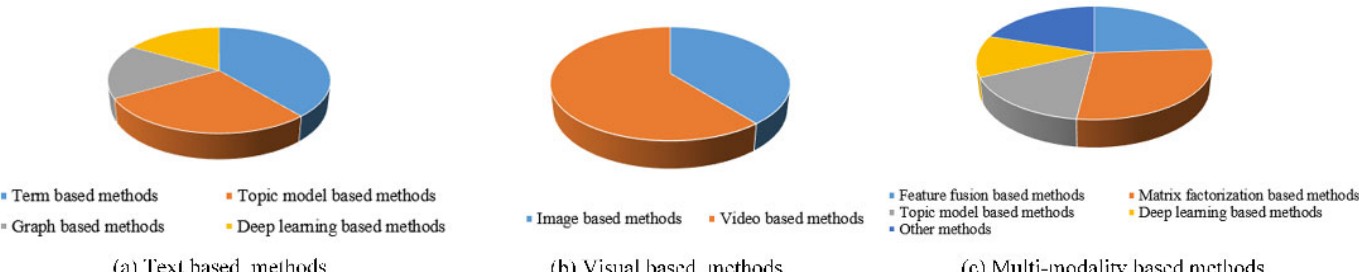

**Figure 2.** Statistics about the different kinds of methods in the literature review.

Figure 2a shows the text based methods, Figure 2b shows the visual based methods, Figure 2c shows the multi-modality based methods. We can observe from Figure 2 that the number of all methods is nearly evenly distributed in our review.

## 3. The Notion of Event Detection and Evolution

There are several important definitions used in this paper. The goal of topic detection is to find, organize and utilize multilingual information from a variety of online data by topic [17]. The concept of "topic" is no longer equivalent to the "topic" in information retrieval, not a certain "field", but a relatively specific "event" [18]. In some cases, "topic" and "event" can be used in general without a strict distinction. Topics can also be thought of as unusual things that happen in a specific time or place. In order to distinguish it

from linguistic concepts, the TDT evaluation conference defined relevant elements [19] as follows:

Topic (definition): Topic consists of a seed event or activity, and all the subsequent events or activities directly related to it. A topic starts with an event $e_1$ and followed by other related events $\{e_2, e_3 \cdots\}$. Each event $e$ contains a set of stories $\{s_1, s_2, \cdots s_i \cdots\}$ and a topic consists of a collection of events $\{e_1, e_2, \cdots e_i \cdots\}$.

Event (definition): Event is special cases caused by specific reasons and conditions, occurring at some special times and places, and may be accompanied by specific consequences. An "event" is different from the definition of "topic". A "topic" can be regarded as a set of events. So, in some special situations, "event" and "topic" can be viewed as the same concept.

Event evolution (definition): The evolution of topics refers to the existence of identity and correlation among topics at adjacent time, which is evolution relationship of these topics with the change of time. The identity of topic is measured by the similarity of topic phrase. The relation between topic and subsequent topic is defined by the semantic correlation degree of topic.

## 4. Single-Modality Data Representation and Event Detection

Data representation is the foundation of event detection because it can translate data into a structural form, which is a kind of computer recognizable information and can be easily understood by an event detection algorithm. Therefore, it is very important to study data representation for event detection. We represent a summary of the data representation in single-modality event detection in this section, including text based methods and visual based methods. We summarize these methods respectively.

### 4.1. Text Based Data Representation

#### 4.1.1. Term Based Methods

In most existing works, text based event detection mainly used the Natural Language Processing (NLP) based methods to represent data. The most widely used representing model is the Vector Space Model (VSM) [20], which is an efficient model and is simple. By VSM, a document is represented by a group of key terms and the corresponding term weights. Words with higher weights are more important to a document [21]. In general, terms can be extracted by Term Frequency-Inverse Document Frequency (TF-IDF) [22], Named Entities Recognition (NER) [23] or Textrank [24]. However, VSM totally ignores words order and words relationship and lack of semantic information. We call this kind of event detection methods the term based methods for that in VSM representation, these terms are isolated. For example, the Group Average Clustering (GAC) proposed by Yang Y et al. [25] used VSM to represent data, in which only the top k terms are selected to represent the document and the rest terms are discarded. The value of k is set by experience and the event detection performance is optimized gradually. Then the similarity measurement is used to measure the documents similarity and vectors are clustered. Finally, the documents are clustered to generate events. The basic idea of term based event detection methods is shown in Figure 3. In addition, the method in [26–30] all used this idea for events detection on document datasets. In these researches, different feature selection and parameter settings may influence the results [31]. So preliminary experiments are required to determine the parameter values.

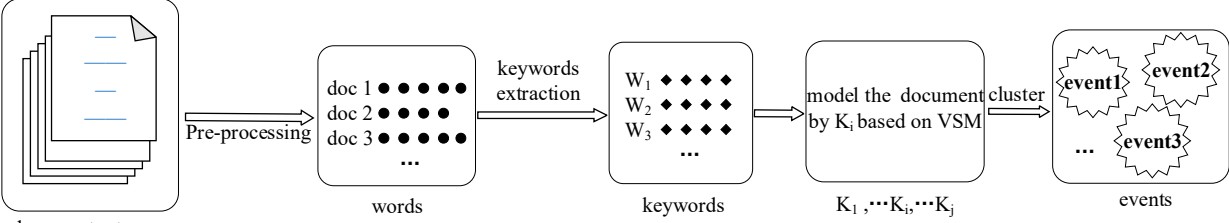

**Figure 3.** The basic idea of term based event detection methods.

### 4.1.2. Topic Model Based Methods

To avoid the above problems caused by VSM representation, topic modeling, such as Latent Dirichlet Allocation (LDA) [32], Latent Semantic Indexing (LSI) [33] and Probabilistic Latent Semantic Indexing (PLSI) [34], is widely used in the next generation for topic (event) detection. In topic modeling, a document is represented as a probability distribution of a set of topics, and each topic is represented as a distribution of a set of topic words. The idea of the topic model is shown in Figure 4, in which $\alpha$ and $\beta$ are the distribution parameters of Dirichlet, w is the word. $\theta$, z, $\varphi$ are the latent variables. K, M, N are the number of topics, number of documents and the number of words in a document, respectively. Topic models is an unsupervised model and it can extract semantic information from text effectively. Topic model had achieved great success on text representation and many researchers had applied topic model on topic detection task. For example, Blei D M and Mcauliffe J D [35] proposed a supervised LDA model that leveraged the documents information to obtain better learning process and achieved better representation of documents. Andrzejewski D et al. [36] proposed to use Dirichlet forest priors to incorporate domain knowledge into topic modeling. In addition, Blei D M et al. [37] used hierarchical topic model to mine topic hierarchies and finer grained topics. Hou L et al. [38] proposed a multifaceted news analysis methods for the online news events search. Li Z et al. [39] provided a new representation of news articles and news events by using probability model that combined content and time information. It can be seen from above researches that the probabilistic topic model method has achieved excellent performance on event detection tasks.

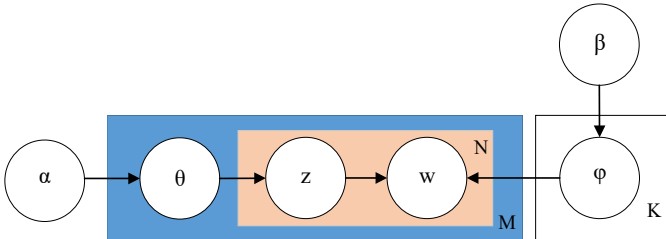

**Figure 4.** The basic idea of LDA.

### 4.1.3. Graph Based Methods

In recent years, some topic detection research based on graph analytical approaches was proposed and achieved excellent performance. In graph analytical approaches, a graph is used to model words co-occurrence frequency in documents and then the graph can be divided into several parts by community detection. Each part represents a topic and the documents in corpus can be assigned to the most related topic. The framework of graph based topic detection is shown in Figure 5. For example, Sayyadi H et al. [2] proposed a topic detection method based on word co-occurrence, which converts documents into graph structure by word co-occurrence between documents, and then used community detection to divide graphs into different parts. Zhang C et al. [40] combined word co-occurrence graph with semantic information graph established by LDA. Their model can emphasize the potential co-occurrence relationship between words in text data, so that important but rare topics can be detected. Chen Q et al. [41] introduced WordNet as external

semantic knowledge in word co-occurrence graph, enriching the semantic information of graph structure. Then they extracted topics from the topic graph by community discovery. They defined a topic pruning process to find the optimal topic with the Markov decision processes. The graph analytical approaches had achieved excellent performance in topic detection task. However, its time complexity is also very high.

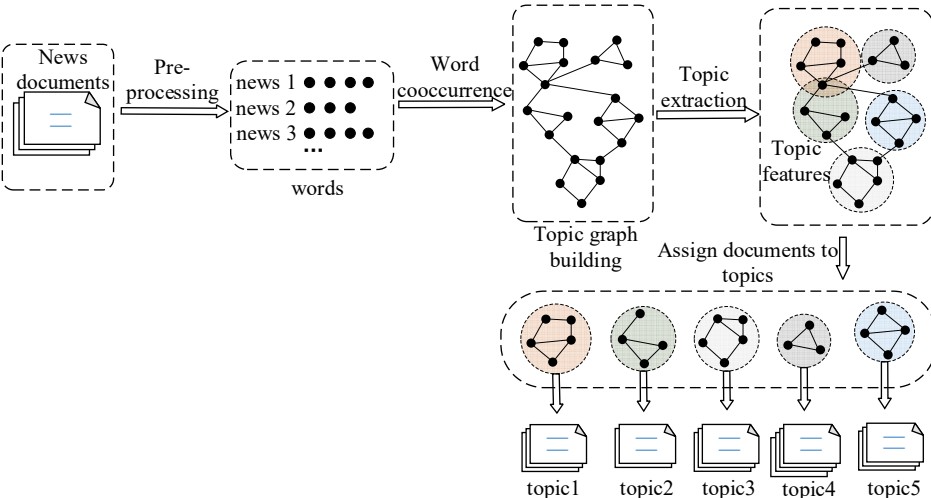

**Figure 5.** The framework of graph based topic detection.

### 4.1.4. Deep Learning Based Methods

With the development of deep learning techniques and natural language processing techniques, deep learning techniques achieved excellent performance on text representation and event detection tasks. The word embedding technique is the first step of data representation based on deep learning. Word embedding techniques learn the representation for each word and words with a similar meaning are close in space [42]. By using word embedding techniques, the problem of sparseness and dimension disaster can be avoided in text representation. Based on word embedding techniques, a series of deep learning based event detection methods have been proposed and achieved excellent performance. For example, Hu L et al. [1] presented a novel text representation method for event detection task based on word embedding. They first transform words into continuous low-dimensional vectors based on word embedding. After that, K-means clustering is used to cluster words that have similar meaning and then a latent semantic space can be obtained. Their proposed representation method can reduce the dimension of document representation and alleviates the sparse semantics, thus improved the efficiency and accuracy of event detection. Hu L et al. [43] also proposed a novel neural model for event detection and prediction, which apply a standard Long Short-Term Memory (LSTM) model [44] to learn the shared event representation between different tasks. The LSTM encoder reads the words within the event sequentially and updates its hidden state iteratively. The method classifies the events to predefined types as well as predicts the next probable event by generating a sequence of words describing it. Bodrunova S et al. [45] proposed a novel approach that incorporates an assessment of the word proximity of texts, combined with text encoding model that is based on the system of sentence embedding. Their approach combines Universal Sentence Encoder (USE) with data pre-processing, agglomerative hierarchical clustering and the Markov stopping moment to optimize clustering. The proposed model performed better than traditional text representation methods.

### 4.2. Visual Based Data Representation and Event Detection

The visual based methods usually leverage the techniques in computer vision field. The visual based methods can be divided into image based methods and video based methods. We discuss the two type of methods separately in this section.

### 4.2.1. Image Based Data Representation

Image is also a common form of data for event detection except for text. There has been a lot of research on event detection based on image information. For example, Imran N et al. [46] proposed a method to classify different event types based on the photos. They mined the most informative features for the event recognition from photo collections. Dense Scale-Invariant Feature Transform (SIFT) [47] and color features are extracted and represented by the Bag-of-Words (BOW) representation. The most important features are selected by PageRank and Support Vector Machines (SVM) and are used to predict the types of events. Bossard L et al. [48] proposed a latent sub-event approach for event recognition. They used discriminative hidden Markov to model the transitions between states, which is called the Stopwatch Hidden Markov model (SHMM). Experiments on the model show that their proposed model outperforms approaches based only on feature pooling or a classical hidden Markov model. Dao M S et al. [49] associated personal image collections with events. They analyzed the photo collection rather than analyzing individual images. They exploited three main features, namely Saliency, GIST and Time to extract an event signature, which is the characteristic for a specific event type. The proposed method detected different event types such as graduation, wedding, or different types of vacations and sports events. Moreover, Ruocco M and Ramampiaro H [50] proposed a method to cluster the images, which takes into account textual annotations, time and geo-location of the images. They also extended a well-known clustering algorithm called Suffix Tree Clustering (STC), which originally developed to cluster text documents using document snippets. Papadopoulos S et al. [51] presented a novel image clustering framework scheme that relies on the creation of two image graphs representing and two kinds of similarity between images, with the similarity based on their visual features and their tags. They use visual and tag similarity and perform clustering on such image similarity graphs by means of community detection. Subsequently, they classify the resulting image clusters as landmarks or events by using features related to the temporal, social, and tag characteristics of image clusters. Cooper M et al. [52] presented a similarity-based method to cluster digital photos, the cluster is based on time and image content.

Guo C et al. [53] proposed a hierarchical model that can be used to recognize events in personal photo collections. The method leverages multiple features such as time, scene and objects. They also study the problem of recognizing events in personal photo albums. An attention network is introduced to learn the representations of photos. Then a hierarchical model is adopted to recognize events from coarse to fine by using multi-granular features [54]. Kaneko T et al. [55] proposed a method to discover events from the Twitter stream. They made use of "geo-photo tweets", which are tweets including both geotags and photos. Their proposed method mined various events visually and geographically by using visual information as well as textual information. Zaharieva M et al. [56] explored a fully automated approach for the detection of specific social events. Both the text and image metadata of photo collection are used. To explore the applicability of visual-based information, they employed visual models which are subsequently used to assign an image to a cluster. Hamrouni A et al. [57] proposed photo-based Mobile Crowdsourcing (MCS) framework for event reporting, which used an A-Tree shape data structure model for clustering streaming pictures to reduce information redundancy and provide maximum event coverage. A summary of the representation of image based event detection is shown in Table 1.

**Table 1.** Representation of image based event detection.

| Reference | Detection Technique | Representation |
|---|---|---|
| [46] | SVM classification | SIFT descriptors, color descriptor |
| [48] | Hidden Markov Model | (1) Global temporal features: time of day, day of week, month; (2) Low-level visual features: densely sampled SURF descriptors; (3) Higher-level visual features: the type of scene, type of indoor scene, number of faces, facial attributes over detected faces. |
| [49] | Classification | Saliency, Gist and Time. |
| [50] | Suffix Tree | Textual annotations, time and geographical positions. |
| [51] | Hybrid image clustering | Visual features (scale invariant feature transform (SIFT)) and their tags |
| [52] | Clustering | Temporal and content (low frequency DCT features) |
| [53] | Coarse classification | Multiple features including time, objects (CNN features) and scenes (CNN feature) |
| [54] | Hierarchical model | Representations based on attention network (including image, attribute, scene and time) |
| [55] | Clustering | Visual information (bag-of-features with densely-sampled SURF local features and 64-dim RGB color histograms) as well as textual information (event keyword detection) |
| [56] | Clustering | Visual dict (used PHOW features to construct a bag-of-visual-words model from the selected image set), text (type, location, time) |
| [57] | Clustering | Combinations of photos visual (SIFT) and semantic features, and the photo proprieties |

### 4.2.2. Video Based Data Representation

Video is also a widely used form of data for event detection tasks. Video can be viewed as an image sequence. Different from images, motion should be considered in the video. Data representation and feature selection of video is of great challenge in video based event detection task. In our review, the video based event detection can be divided into three categories: low level feature based methods, high level feature based methods and deep learning based methods.

Low level features can be divided into static frame based visual features and motion visual features. In which, static frame based visual features are the most widely used in the video event detection. The most known static frame based features is Scale-Invariant Feature Transform (SIFT) [47] that proposed by Lowe D G, and it has been successfully used in some event detection works. For example, to combine the features extracted from video for complex event recognition, Tang K et al. [58] proposed a method that can be selective of different subsets of features including SIFT and the sets of features are considered independently. Some researchers used the improvements of SIFT, such as colorSIFT [59], for video event detection task. For example, Lan Z et al. [60] used three image features including are SIFT, ColorSIFT and Transformed Color Histogram (TCH) for the video detection task.

There are some events that evolve with time, which are more complex than static events. To deal with this situation, the motion visual features have been proposed. For example, Tamrakar A et al. [61] used both the static and dynamic features including the Motion SIFT (MoSIFT) [62], which is a 3D version of SIFT, for complex event detection. In addition, Yang Y. [63] used three motion features including Spatio-temporal interest points (STIP) [64], Motion SIFT and Dense Trajectories for complex event detection. Tang K et al. [58] extracted the 3D Histogram of Oriented Gradients (HoG3D) [65] and computed video-level histograms for complex event recognition. Alamuru S and Jain S [66] detected the multiple events in the videos, in which the gradient local ternary pattern, histogram of oriented gradients, and Tamura features are used. This feature is extracted from the enhanced frames.

Researchers have also studied high level representation for video based event detection. For example, Merler M et al. [67] proposed semantic model vector to detect events in unconstrained videos. The semantic model vector is an intermediate level semantic representation and is extracted using a set of discriminative semantic classifiers, each being an ensemble of SVM models. An end-to-end video event detection system was adopted and combined semantic model vectors with other static or dynamic visual descriptors, extracted at the frame, segment, or full clip level. Gkalelis N [68] proposed to use a model vector-based approach, where visual concept detectors are used to automatically describe a video sequence in a concept space. In addition, there are some researchers who used high-level video representations with a set of low-level features for video event detection. For example, Yu Q et al. [69] used the static features, dynamic features and the bag-of-word representations to represent action, scene and object concepts.

With the development of deep learning, many researchers used deep convolutional neural networks (CNN) for computer visual tasks and has achieved great success in video processing [70]. For example, Xu Z et al. [71] were the first to propose to use encoding techniques for video representation generation. Their proposed method is a discriminative video representation on large dataset under limited hardware resources. The representation effectively applied deep Convolutional Neural Networks (CNNs) to the event detection task. To enrich visual information, they proposed to use a set of latent concept descriptors as the frame descriptor. Zha S et al. [72] surveyed different methods for event detection and action recognition in videos by using convolutional neural networks (CNN). Results show that the CNN-based approach can outperform the static and motion-based Fisher vector techniques. What is more, integrating motion information with simple late fusion can improve classification performance. Ye G et al. [73] proposed a large scale event-specific concept library that covers many real-world events and concepts, which is called EventNet. They trained a Convolutional Neural Network (CNN) model to extract deep learning feature from video content. Tian H et al. [74] proposed a new multimodal deep learning framework for video event detection, which can automatically generate deep features from each modality.

Besides the CNN, there also some recent works that used other deep learning methods. For example, Pouyanfar S et al. [75] proposed a new ensemble deep learning framework, which can be utilized in various scenarios and datasets. Their proposed method not only overcomes the imbalanced data issue, but also decreases the information loss and overfitting problems caused by single models. Xu H et al. [76] proposed the Joint Event Detection and Description Network (JEDDi-Net). The model utilized three-dimensional convolution to extract video appearance and motion features, which are sequentially passed to the temporal event proposal network and the captioning network. Zhang L et al. [77] proposed a two stage neural network for the video event classification task, where the first stage can transfer pre-learned object knowledge to video contents and the second stage can combine temporal information by RNN. A summary of the representation of video based event detection methods is shown in Table 2.

**Table 2.** Representation of video based event detection.

| Category | | Reference | Representation |
|---|---|---|---|
| Low-level feature | Static frame based visual features | [58] | Various static features including Scale-Invariant Feature Transform (SIFT) |
| | | [60] | Three image features including SIFT, ColorSIFT and Transformed Color Histogram (TCH) |
| | Motion visual features | [61] | Motion SIFT (MoSIFT) |
| | | [64] | Spatio-temporal interest points (STIP), Motion SIFT and Dense Trajectories |
| | | [58] | Various features including 3D Histogram of Oriented Gradients (HoG3D) |
| | | [66] | Gradient local ternary pattern, histogram of oriented gradients and Tamura features |
| High level representation | | [67] | Semantic model vectors which is an intermediate level semantic representation extracted using a set of discriminative semantic classifiers. |
| | | [68] | Decomposed video to a sequence of shots and trained visual concept detectors are used to represent video content with model vector sequences. |
| | | [69] | Static features (i.e., SIFT) and dynamic features (i.e., STIP and Dense Trajectory Based features) |
| Deep learning based methods | | [70] | Deep convolutional neural networks (CNN) |
| | | [71] | Deep convolutional neural networks (CNN) |
| | | [72] | Deep convolutional neural networks (CNN) |
| | | [73] | Deep convolutional neural networks (CNN) |
| | | [74] | Deep convolutional neural networks (CNN) |
| | | [75] | Enhanced Ensemble Deep Learning |
| | | [76] | Three-dimensional convolutional layers |
| | | [77] | Two-stage neural network strategy |

## 5. Multi-Modality Data Representation and Event Detection

As the various online media applications proliferate, a large amount of multi-modality content is available on the internet. This content is being created and is massive. Thus, detecting events from the multi-modality data is of great significance. A multi-modality event detection system can detect events and group the related data by the representation of event data. It requires the analysis of large scale data from different sources [78], so it is a great challenge to find events from the online multi-modality data. In multi-modality event detection, the data representation is the most important part. The framework of multi-modality data representation and event detection is shown in Figure 6. The basic idea is to project multiple types of modality data in a commonly shared space to make sure that the different modalities can be compared.

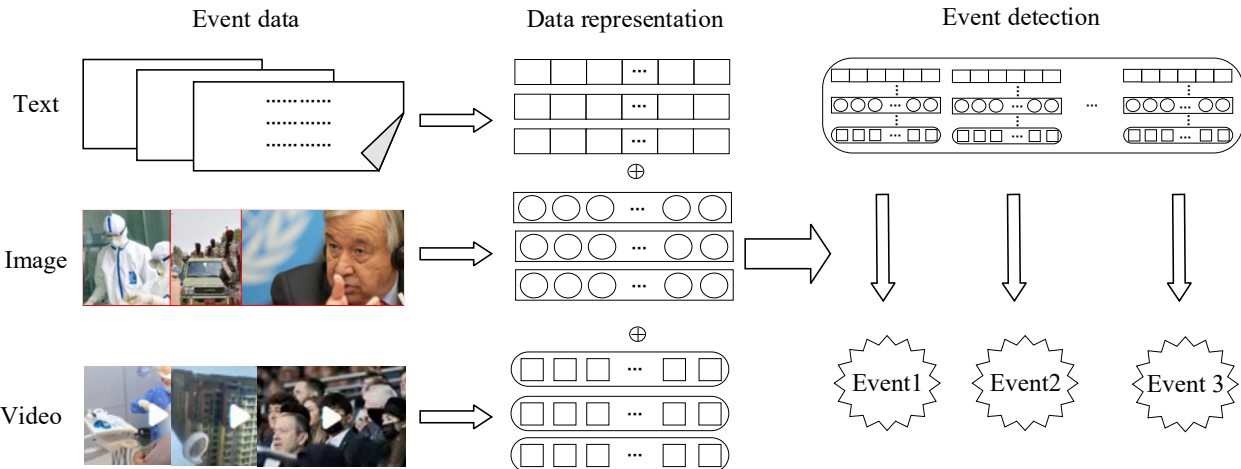

**Figure 6.** Framework of multi-modality data representation event detection.

Multi-modality event detection can provide more comprehensive understanding of events. Multi-modality data have brought new opportunities for people to scene what is happening in the world and provide a great challenge to researchers. Firstly, the multi-modality data are heterogeneous and it is very hard to analyze the data, the data are from different sources in multiple modalities. In addition, the multi-modality data are of multiple characteristics and attributes. The multiple characteristics and attributes are also heterogeneous and important for events detection. Therefore, the representation of the multi-modality data is of great significance for multi-modality event detection. Many researchers have made a deal of efforts to address these challenges. In this section, we classify these methods into feature fusion methods, matrix factorization based methods, topic model based methods, deep learning based methods and other methods.

### 5.1. Feature Fusion Based Methods

Event detection from multi-modality data is a difficult task that combines text, images and video. To exploit the heterogeneous data and multiple feature modalities, different fusion approaches including early fusion and late fusion have been proposed. In early fusion, also known as feature level fusion, all the features including textual and visual features are concatenated into a single feature vector at feature level [79]. For multi-modality event detection with early fusion, a set of concatenated features is feed into a classifier for training. Thus, the trained classifier can separate test data into different event types. Late fusion combined the posterior probabilities obtained from the classifiers of each feature type. In this case, several models are trained for text, image and video and then combined at decision level [80]. Late fusion combines the confidence scores of different models by leverage feature modalities. The confidence score measures the probability of classifying the data into different event types. Different to early fusion, late fusion trained separate classifiers for the features and then several classifier outputs probabilities for the different types instead of predicted labels.

In addition, the intermediate fusion methods are proposed. For example, some works represent different modalities and fuse them by extracting joint latent topics using a multi-modal extension of LDA [81]. Mei T et al. [82] presents a novel probabilistic approach which can fuse multimodal metadata for event based home photo clustering. They incorporated these multimodal metadata into a unified probabilistic framework. Some more complex fusion methods are also proposed. These models used more learning stages to optimize the weights of each feature modality to the final fusion event detection models. In addition, many works have constructed event detection models by combining the early, late, and other fusion techniques [83,84].

## 5.2. Matrix Factorization Based Methods

Matrix factorization based methods aim to learn low-dimensional representation of data for event detection. However, the constraints such as non-negativity, sparsity, low-rank, and so forth, should be considered by observing the data characteristics. The constraints have been widely used and research shows that they are effective in feature extraction and dimension reduction. Matrix decomposition aims to decompose a matrix into two sub-matrices. The idea of matrix factorization can be used in multi-modality representation learning without considering the number of modalities. Recently, some event detection research explored multimodal data representations. Representative works include Singular Value Decomposition (SVD) [85], Non-negative Matrix Factorization (NMF) [86,87] and Dictionary Learning (DL) [88,89]. For example, Xue Z et al. [90] proposed a semi-supervised co-clustering method for multi-modality topic detection. In their work, the non-negative matrices are used to represent multi-modality data, in which the matrix represents the relationship between the central data and the feature modality. Then the clusters of the central data and every feature modality can be obtained by tri-factorizing these matrices simultaneously, and every cluster of the central data can correspond to an event. Gupta S K et al. [86] proposed a shared subspace learning framework, in which a subspace shared between different data sources is learned by a Nonnegative Matrix Factorization. The proposed method learns co-occurrences from the subspaces of the target and auxiliary datasets by explicitly learning a common subset of basis vectors. There are also some studies proposing the multimodal graph regularized sparse representation methods, which are achieved by concatenating the unimodal data representations. For instance, Yu Z et al. [91] proposed a discriminative coupled dictionary hashing (DCDH) method, in which the side information is used for the learning of coupled dictionary for each modality. The coupled dictionaries can be used to preserve the intra-similarity and inter-correlation among multi-modality data, and it also contain dictionary atoms that are semantically discriminative. Yu J et al. [92] proposed a multimodal hypergraph learning-based sparse coding method by concatenating the unimodal data representations. They used the method to predict image click and applied the click data to the reranking of images. Sharma A et al. [93] presented a multi-view feature extraction approach called Generalized Multiview Analysis (GMA). GMA has various important properties that are required for cross-view classification, which is a supervised model relying on labeled data for training.

## 5.3. Topic Model Based Methods

Probabilistic models have been widely used in event detection. However, most probabilistic model based research detects events in documents. In these methods, each document is represented as the distribution on a set of topics, and each topic is represented as the distribution on a set of topic words, as discussed in Section 4.1.2. For multi-modality event detection, due to the fact that the multi-modality data not only contain texts, but also contain visual information such as image and video. Therefore, topic models for multi-modality event detection had been researched by some researchers.

Several methods have been proposed to apply topic models to deal with the multi-modality data [94,95]. In multi-modality data based event detection, not only words but also other modalities need to be taken into consideration. Cai H et al. [95] proposed a novel topic model that models the text, image, location, timestamp and hashtag feature of Twitter to discover events. In this model, each tweet is assigned to a topic and each topic has a mixture of several important Twitter features distributions. Qian et al. [96] proposed a boosted multimodal supervised Latent Dirichlet Allocation (BMM-SLDA), which is used for the event classification task. The model integrated the multi-modal supervised Latent Dirichlet Allocation (mm-SLDA) [97] in the boosting framework. The BMM-SLDA can effectively exploit the multi-modality and the multiclass property of social events, as well as make use of the label information to classify multiclass social event directly. Qian et al. [98] also proposed a novel multi-modal event topic model (mmETM), which can effectively model social media information such as text with images. It also learns the correlations

between textual and visual modalities to separate the visual-representative topics and non-visual-representative topics.

### 5.4. Deep Learning Based Methods

Some works also proposed deep learning based multi-modality event detection. For instance, Chang X et al. [99] proposed a bi-level semantic representation analyzing method. Their method learns semantic representation weights, and the negative influence caused by the noisy or irrelevant concepts are restrained. Lv J et al. [100] presented a novel heterogeneous graph embedding method HGE2MED, which can be used to learn the heterogeneous relations and obtain the representation of each node. They obtained heterogeneous walking sequences by heterogeneous random walk, which can mine the relation of graph structure and triplet sample and outperforms low-rank representation. Abebe M A et al. [101] introduced a generic Social-based Event Detection framework (SEDDaL), the model input a collection of social media objects from heterogeneous sources and produced semantically meaningful events interconnected with spatial, temporal, and semantic relationships.

### 5.5. Other Methods

There are also some other data representation methods for multi-modality event detection, which cannot be categorized into the above categories. For instance, Yang Z et al. [102] present a shared multi-view data representation (SMDR) model for multi-domain event detection that learns the intrinsic structures between different data views. Huang P Y et al. [103] combined the textual and visual information by nonlinear Canonical Correlation Analysis (CCA), in which the LSTM is used to pre-encode the textual and visual features and multi-layer perceptron with a residual link. There are also some methods based on hashing that obtain features of multi-modal data. In these methods, different modalities are projected into a commonly shared space. For example, Donald Metzler D et al. [104] proposed a method to retrieve a list of historical event summaries, in which high quality event representations is extracted by using the temporal query expansion technique. Li W et al. [105] proposed a multimodal topic and-or graph (MT-AOG), which can be used to represent important textual and visual elements of news stories. MT-AOG models latent topic structures by leveraging a context sensitive grammar that can describe the hierarchical composition of news topics by semantic elements about people involved, related places, and what happened.

The advantages and disadvantages of the multi-modality event detection methods are shown in Table 3.

It can be seen from Table 3 that all models have some advantages, but the disadvantages of some models are not introduced, mainly because most of these models are applied to specific scenarios and solve specific application problems. Since the researches in the field of multi-modality event detection faces different scenarios and the problems to be solved are also different, these models are not universal. Most of the researches are carried out on the self-collected datasets, so it is difficult to evaluate which model is the best or worst. Only a small number of studies were evaluated under the same dataset, and their performances were compared in the evaluation part of Section 7.

**Table 3.** Advantages and disadvantages of the multi-modality event detection methods.

| Category | Reference | Advantages | Disadvantages |
|---|---|---|---|
| Feature fusion based methods | [79] | Utilize various features | Lack of social network information |
| | [80] | Constrained clustering algorithm is used to achieved high accuracy | Imbalanced data and parameter setting is not optimal |
| | [81] | Automatic concept mining and boosted concept learning | Application is limited |
| | [82] | Unsupervised and without predefined threshold | Lack of more types of semantic features |
| | [83] | Need no manual annotation and can adapt concepts to news domains | - |
| | [84] | Utilized the information contained in the related exemplars | - |
| Matrix factorization based methods | [85] | Robust to data incompleteness | - |
| | [86] | Can discover the shared structure between the datasets | - |
| | [90] | semi-supervised co-clustering with side information | Parameters setting is not automatic |
| | [91] | Contain dictionary atoms that are semantically discriminative | - |
| | [92] | Predict image clicks and solved the problem of lack of data | - |
| Topic model based methods | [94] | Generate visualized summaries | Lack of personalized microblog summarization |
| | [96] | exploit the multimodality and suitable for large-scale data | Without videos and audios modality |
| | [97] | Exploit various property jointly and classify multi-class events | - |
| | [98] | Can classify the visual-representative topics from non-visual-representative topics | Didn't consider different domains |
| Deep learning based methods | [99] | restrains the negative influence of noisy or irrelevant concepts | - |
| | [100] | Maintain multi-view information by robust representation | - |
| | [101] | A generic model to describe events and their relationships | - |
| Other Methods | [102] | Deal with multi-view tasks | Poor data quality and high complexity |
| | [103] | Jointly regularizing the encoded representations | Lack of event summarization and itemization |
| | [104] | Retrieval structured event representation, robust and effective | - |
| | [105] | Structured topic representation | - |

## 6. Multi-Modality Event Evolution

The multi-modality events evolution analysis trains a model based on the event multi-modality data. Event evolution analysis aims to find the correlation among topics at the adjacent time, which is an evolution of the relationship of these topics with the change of time. Event evolution analysis can help the user to understand the development or trend of an event.

There are many works researching how to model the event evolution within multi-modality data. Previous evolution modeling methods are mainly focused on a single modality such as text and video/image. However, multi-modality data are very different from single modality data. So, the evolution of multi-modality event evolution is a great challenge and needs completely different methods. Neo S Y et al. [106] proposed to improve news video search by using the semantics of video and relevant external information resources, and then discover topic hierarchy for browsing key events and supporting question answering (QA). The work combined multimodal event information extracted from various online platforms for event evolution analysis and introduced topic evolution based on the interest of users. What is more, they extended QA to deal with the various types of specialized video queries. To develop the browsing systems that can allow us to search results with rich information mined from various sources, the synchronization of multiple media content is investigated in the form of hyperlinks [107]. This method can achieve the visualization and exploration of different information landscapes inherent in search results. The content mining and selection from web videos, space-time alignment of multiple media, and augmenting of search result with when and what information are studied in the work for developing these browsing features. Wu X et al. [108] explored to discover event from web video and model the event evolution structures. Web users can better understand the major event through a concise structure that shows the events evolution associated with the representative text keywords and visual shots. The event structure represents the important properties, which can realize video visualization and browsing effectively. Wang D et al. [109] introduced a novel framework that used text and image data to generate storylines for a given topic. The key idea is to construct a weighted multi-view graph that can be used to capture the context and temporal relationships between the topic-related objects, given a set of images and their textual descriptions. Then the objects are selected by solving the minimum weight connected dominant set problem defined on the graph. The proposed framework provides a storyline with text, images, and structural information, and provides sketches of the topic evolution. Shan D et al. [110] proposed a system for event extraction and retrieval called EventSearch. The system was conducted on several types of historical data such as news articles, newspapers, TV news, and micro-blog messages. The system detects events and can provide a better understanding of event evaluation and causalities among events. The system also provides a visualization for events based on multimedia, in which each event snippet consists of multi-modality data elements. In order to enhance the readability of information and improve user utility, different types of data are combined by a systematic manner. Xu S et al. [111] proposed a novel solution to extract and reconstruct the storylines. Specifically, they first investigated the requisite properties of the storyline and devised an algorithm to extract all effective storylines, in which the properties are optimized at the same time. Finally, the extracted lines are all reconstruct and generated the story map with high quality. Qian S et al. [98] proposed a multi-modal event topic model (mmETM) for the multi-modal topics detection and evolutionary trends analysis, which can be used to generate event summary effectively. mmETM models social media data including the text and their related images. For social event tracking, they use the mmETM to obtain the textual and visual topics of social events. Qian S et al. [112] later proposed a nonparametric tracking model for event tracking, which is an online multimodal multi-expert learning algorithm that can automatically learn the number of topics. It adopts a novel multi-expert minimization restoration scheme. The model drift problem in event tracking can be alleviated by allowing the tracked model to evolve backward and canceling unnecessary model updates. Li W et al. [106] proposed multimodal topic and-or graph (MT-AOG) that can represent textual and visual information and their latent topic structures. The model can track and update the incoming news streams. The topics detected in different time periods are linked and topic trajectories can be generated, which shows how topics emerge, evolve, and disappear over time. The MT-AOG model can also effectively track and update the states of news. Table 4 shows the comparison among typical multi-modality event evolution models.

**Table 4.** Comparison of the multi-modality event evolution models.

| Models | Modality Data Type | Characteristic |
|--------|--------------------|----------------|
| [107] | News video, web news articles and news blogs | Topic evolution based on various online platforms and based on the interest of users. |
| [108] | Web video, news article | Topic evolution with various media information, and enables the visualization and exploration of different information. |
| [109] | Web video | A concise structure that shows the evolution of events associated with the representative text keywords and visual shots. |
| [110] | Images, text | Text and image are combined to analysis and deliver the storyline and structural information to provide topic evolution sketch. |
| [111] | Web news, newspaper, TV program, Weibo | Events in a very long time period are detected and traced, and a multimedia-based visualization is provided. |
| [112] | Web news with image | Storyline extraction and reconstruction, a unified algorithm is designed to extract all effective storylines. |
| [99] | Web news, image | An incremental learning strategy is adopting informative textual and visual topics of social events over time to help understand events and their evolutionary trends. |
| [106] | News Text, image | Tracking topics detected in certain continuous time periods. Link all detected topics in different time periods to form topic trajectories over time. |
| [113] | Web news, images | Nonparametric online multimodal tracking module that allows the tracked model to evolve backwards to undo undesirable model updates, which helps alleviate the model drift problem of social event tracking. |

## 7. Datasets and Evaluation

### 7.1. Datasets

Most of the existing works used self-collected datasets and these datasets are not public. So, it is hard to compare their performance. However, there are also some public datasets were used to evaluate multi-modality event detection. So, we only give the comparison between some specific works on several public datasets in this section.

(1) Social Event Detection (SED) dataset. The SED datasets are provided by the MediaEval challenge. The goal of this task is to discover social events of interest from the mass of user-generated Flickr multi-modality content and the metadata surrounding it. It released four subsets in 2011–2014: SED2011 [113], SED2012 [114], SED2013 [115], SED2014 [116]. Different years released different challenges. The statistic information of the datasets is shown in Table 5.

**Table 5.** Datasets of SED 2011~2014 for event detection.

| Data | Time | Introduction |
|------|------|-------------|
| SED2011 | 2011 | Contains 73,645 images of five cities: Amsterdam, Barcelona, London, Paris and Rome. |
| SED2012 | 2012 | Contains 167,332 images of five cities: Barcelona, Madrid, Cologne, Hamburg and Hannover |
| SED2013 | 2013 | Contains 427,370 images and 1327 videos with XML timestamps, geographic information, tags, titles, descriptions, etc. |
| SED2014 | 2014 | SED 2014 released two datasets. One dataset contains 362,578 images and the other dataset contains 110,541 images. The metadata of the two datasets are available. |

(2) Multi-modality Social Event Dataset. This dataset is a multi-modality social event dataset (short for MMSE) downloaded from Flickr and has different versions. The datasets are composed of image and text data. The first version was released in 2014 and contains 36,000 documents that consist of 10 types of events [97], the second version was also released in 2014 and contains 59,500 documents that consist of 12 types of events [117], the third version was released in 2015 and contains 107,600 documents that consist of 18 types of events [96], the latest version named HFUT-mmdata was released in 2019 and contains 74,364 documents that consist of 10 types of events [118]. The details of these datasets are shown in Table 6.

**Table 6.** Multi-modality social event dataset based on Flickr.

| Dataset Name | Source | Number of Event Types | Content | Documents Number of Each Event | Total Documents Number | Year |
|---|---|---|---|---|---|---|
| MMSE (version 1) | Flickr | 10 | image and text | 2500~5000 | 36,000 | 2014 |
| MMSE (version 2) | Flickr | 12 | image and text | 3000~6000 | 59,500 | 2014 |
| MMSE (version 3) | Flickr | 18 | image and text | 4000~8000 | 107,600 | 2015 |
| HFUT-mmdata | Flickr | 10 | image and text | 7000~9000 | 74,364 | 2019 |

### 7.2. Evaluation

Table 7 shows the comparison of results of SED 2011~2014 under different challenges. The F-measures as well as the normalized mutual information (NMI) are used to evaluate the performance.

**Table 7.** Results of comparison results on SED 2011–2014 datasets.

| | | Challenge 1 | | Challenge 2 | | Challenge 3 | |
|---|---|---|---|---|---|---|---|
| | | **F-Score** | **NMI** | **F-Score** | **NMI** | **F-Score** | **NMI** |
| SED2011 | [119] | 0.69 | 0.41 | 0.33 | 0.54 | | |
| | [120] | **0.77** | **0.63** | 0.64 | 0.38 | | |
| | [121] | 0.59 | 0.27 | **0.69** | **0.62** | | |
| | [122] | 0.65 | 0.24 | 0.50 | 0.45 | | |
| SED2012 | [123] | 0.22 | 0.02 | 0.30 | 0.20 | 0.48 | 0.31 |
| | [124] | **0.85** | **0.72** | **0.91** | **0.85** | **0.90** | **0.74** |
| | [125] | 0.19 | 0.18 | 0.75 | 0.67 | 0.67 | 0.47 |
| | [126] | 0.70 | 0.60 | NA | NA | 0.61 | 0.45 |
| SED2013 | [127] | 0.57 | 0.87 | | | | |
| | [128] | **0.95** | **0.99** | | | | |
| | [129] | 0.88 | 0.97 | | | | |
| | [130] | 0.93 | 0.98 | | | | |
| | [131] | 0.88 | 0.97 | | | | |
| | [132] | 0.78 | 0.94 | | | | |
| SED2014 | [133] | NA | 0.98 | | | | |
| | [134] | 0.97 | 0.99 | | | | |
| | [135] | 0.94 | 0.98 | | | | |

The results of the four years are shown in Table 7. In which, the bold number represents the best results of different challenges under SED2011–SED2014. In the first year (2011), [120] achieved the best performance in challenge 1, mainly because it classifies the photos to cities at first and then partitioned the photos into buckets that contains the photos of the same day or same city. In challenge 2, [121] achieved the best performance, mainly because the approach matches the photos to event descriptions retrieved from online event directories.

In the second year (2012), the performance of challenge 1 is worse than the performance of challenge 2 and challenge 3, it is because the term "technical events" is fuzzy. The

performance of challenge 2 is better than the performance of challenge 3, it is because soccer events are much clearer than the Indignados events. Ref. [124] achieved the best performance on the three challenges. Because it also contains a city classification step and then LDA is used for topic detection for each city.

In the third year (2013), the objectives of challenge 1 has some difference with SED 2011 and SED 2012, and no filtering step was required for all images in SED 2013. Since the photos in dataset were related to metadata, this actually involves a multimodal clustering. Ref. [128] achieved the best performance, because that the challenge of clustering is easier than challenges of SED 2012 and the additional process of filtering or classification is not required.

In the fourth year (2014), the three methods achieved similar performance. SED 2013 and SED 2014 performed better than SED 2011 and SED 2012, mainly because in SED 2013 and SED 2014 datasets, the non-events data are filtered in advance. Therefore, filter non-event data in advance can effectively improve the event detection performance. From the results, we can see that, there is still space for improvements of the multi-modality event detection tasks.

As shown in Table 8, seven methods are compared and evaluated. In which SLDA (visual) and SLDA (text) are the supervised Latent Dirichlet Allocation (SLDA) [136] with only visual features and textual features, respectively. mmLDA + SG and mmLDA + SVM are multi-modal Latent Dirichlet Allocation (mmLDA) [137] using softmax regression method and Support Vector Machine (SVM) for classification, respectively. mm-SLDA [97] is a supervised version of mm-LDA [137]. BMM-SLDA [117] is boosted multi-modal supervised Latent Dirichlet Allocation that used boosting weighted sampling strategy to train the corresponding topic models. KGE-MMSLDA [118] is the knowledge priors- and max-margin-based topic model that integrates additional knowledge from an external knowledge base.

**Table 8.** The results comparison on multi-modality social event dataset.

| | SLAD (Visual) | SLDA (Text) | mmLDA + SG | mmLDA + SVM | mm-SLDA | BMM-SLDA | KGE-MMSLDA |
|---|---|---|---|---|---|---|---|
| MMSE (version 1) | 0.359 | 0.758 | 0.699 | 0.755 | **0.803** | NA | |
| MMSE (version 2) | 0.401 | 0.717 | 0.671 | 0.715 | 0.766 | **0.877** | |
| MMSE (version 3) | 0.312 | 0.702 | 0.665 | 0.724 | 0.722 | **0.835** | |
| HFUT-mmdata | | | | | 0.763 | | **0.851** |

Table 8 shows the performance on multi-modality event datasets (MMSE and HUFT-mmdata) and the accuracy values. In which, the bold number represents the best results of methods under different multi-modality social event dataset. It is obvious that the performance of SLDA (text) is much better than that of SLDA (visual) on all datasets, which means that the textual information is more important than visual information on multi-modality event detection tasks. In addition, SLDA (Text) is better than mmLDA (SG); this is because mmLDA only adopts the multi-modality information and SLDA uses the supervised information. It means that supervised information is useful. However, multi-modality based methods, such as mm-SLDA, BMM-SLDA and KGE-MMSLDA, achieved better results compared with the single-modality based methods such as SLDA (visual) and SLDA (text), which means that multi-modality feature is effective for event detection. In addition, results on HUFT-mmdata show that KGE-MMSLDA performs the best compared with other methods. It is because KGE-MMSLDA integrates additional knowledge into a unified topic model, which means that embedding external knowledge can effectively improve the event detection performance.

## 8. Conclusions

In this paper, we give a comprehensive review of multi-modality event detection. We first reviewed the related definition and then discussed the various event representation

approaches including single-modality data representation and multi-modality data representation for event detection. We discussed the single-modality event representation in different media types, that is, text based methods and visual based methods. For the former, we discussed various feature representation methods in detail, such as term based methods, topic model based methods, graph based methods and deep learning based methods. For the latter, we discussed in detail the image based methods and video based methods. Then, we introduced multi-modality data representation for event detection, we classified the methods into feature fusion based methods, matrix factorization based methods, topic model based methods, deep learning based methods and other methods. For each kind of method, we introduced the representative methods and their characteristics. Based on this, we further introduced the multi-modality event evolution methods. Subsequently, we summarized the most popular public datasets for multi-modality event detection and compared the results of representative works on these datasets. Results show that multi-modality data perform better than single-modality data.

**9. Future Work**

With the rapid growth of multi-modality data, online data are now considered to be more complex with the characteristic of multi-modality, cross-platform, noisy and information redundancy. Data representation became an important step of event detection tasks. Although some significant achievements have been made in this field, multi-modality event detection remains a very difficult task and further efforts can be made in the following aspects in the future.

(1) Take advantage of the diversity of multi-modality data. Multi-modality data types are not limited to modality diversity, such as textual and visual information, but also include social links and user behavior information such as comments and repost. Therefore, we should consider how to use this information in future research;

(2) Event evolution problems. Although there have been many topic evolution studies on multi-modality data, which have yielded valuable results, there are still some problems such as no standard model evaluation benchmark for multi-modality topic evolution models. Therefore, an evaluation method for multi-modality topic evolution models should be proposed for the performance evaluation;

(3) Lack of public datasets. Although there are some public datasets for multi-modality event detection, such as SED 2011–SED 2014 and multi-modality social event datasets mentioned above, these datasets still lack diversity. Therefore, it is necessary to develop more diversified datasets that contains more attribute information and conduct more comprehensive multi-modality event detection in the future;

(4) Improve the interpretability of event detection models. The current event detection models are mostly black-box models. The model and results lack reasonable explanation. Therefore, we should focus on how to improve the interpretability of event detection models in the future;

(5) Improve the data representation for multi-modality data. In existing multi-modality topic detection methods, modality fusion is mostly based on simple feature concatenation. In addition, time synchronization in different modalities is not considered. What is more, existing representations also have high time complexity, which is what we need to research in the future.

**Author Contributions:** Investigation, K.X. and B.Q.; resources, K.X. and B.Q.; writing—original draft preparation, K.X. and Z.Q.; rewriting and editing, Z.Q. and B.Q.; supervision, Z.Q. and B.Q. All authors have read and agreed to the published version of the manuscript.

**Funding:** This research is funded by Humanity and Social Science Youth Foundation of Ministry of Education of China under grant number 21YJCZH117; National Natural Science Foundation of China under Grant number 61772534 and Open Project Program of National Engineering Laboratory for Agri-product Quality Traceability under grant number AQT-2018-YB4.

**Institutional Review Board Statement:** Not applicable.

**Informed Consent Statement:** Not applicable.

**Conflicts of Interest:** The authors declare no conflict of interest.

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
