# Peer review of "A Survey of Data Representation for Multi-Modality Event Detection and Evolution"

_applsci, doi:10.3390/app12042204_

Round 1

Reviewer 1 Report

This paper has the potential to be accepted, but there is a point that has to be clarified or fixed.

If they can summarize the Conclusions and Future work in different ways. It would be better to divide it into two sections.

One section is for Conclusions and the second one is for  Future work.

Author Response

Review 1:This paper has the potential to be accepted, but there is a point that has to be clarified or fixed. If they can summarize the Conclusions and Future work in different ways. It would be better to divide it into two sections. One section is for Conclusions and the second one is for Future work.

Response1: Thank you very much for your suggestion. We have divided the conclusion and future into two parts, Section 8 is the conclusion section and Section 9 is the future work section, as shown in the revised manuscript. Your suggestion makes the structure of our manuscript much better than before. Thank you very much.

Reviewer 2 Report

In this work, authors present a comprehensive review of multi-modality event detection accompanied by many bibliographical references.

It is a particularly exhaustive work and may be useful for many researchers in this area, especially for those who are about to begin their research in this field, this being its greatest contribution.
The document is very well organised and written for its intended purpose.

Author Response

Reviewer2:In this work, authors present a comprehensive review of multi-modality event detection accompanied by many bibliographical references.

It is a particularly exhaustive work and may be useful for many researchers in this area, especially for those who are about to begin their research in this field, this being its greatest contribution. The document is very well organized and written for its intended purpose.

Response2: Thanks very much for your agreement. We are very grateful for your professional comments.

Reviewer 3 Report

The authors present a review on data representation for multi-modality event detection. The article seems well organized. However, I consider that vital information is missing that the authors should add or justify why it was not added.

1) What systematic review methodology did you use? PRISM?
2) What years does the review cover?
3) What databases were consulted? IPDM? IEEE EXPLORE? SCIENCE DIRECT?
4) What were the search keywords?
5) How many articles were included? Which ones were omitted and why?
6) Why didn't you add a bibliometric analysis?

Author Response

Reviewer 3:The authors present a review on data representation for multi-modality event detection. The article seems well organized. However, I consider that vital information is missing that the authors should add or justify why it was not added.

1) What systematic review methodology did you use? PRISM?

2) What years does the review cover?

3) What databases were consulted? IPDM? IEEE EXPLORE? SCIENCE DIRECT? 4) What were the search keywords?

5) How many articles were included? Which ones were omitted and why?

6) Why didn't you add a bibliometric analysis?

Response3: Thanks very much for your suggestions, which are very helpful for us to improve our manuscript. We have added the Section 2 to introduce the methodology under the PRISMA guidelines. The detailed information are show as the yellow highlighted part in Section 2 in our revised manuscript. In addition, a bibliometric analysis is also added to analyze the literature, as shown in Figure 1 in section 2.2.

Reviewer 4 Report

This paper comprehensively review the existing researches of event detection and evolution. 

I would recommend to add table with list of advantages and disadvantages of each model.

In addition, the main issues and challanges for future research direction should be clear.

Which model is the best and what model is the worest? Please jutify your choice

What did you learn from survey event detection?

Author Response

Reviewer 4:This paper comprehensively review the existing researches of event detection and evolution. I would recommend to add table with list of advantages and disadvantages of each model. In addition, the main issues and challenges for future research direction should be clear. Which model is the best and what model is the worest? Please jutify your choice. What did you learn from survey event detection?

Response4: Thanks very much for your suggestions, which are very helpful for us to improve our manuscript. We have added the Table 3 that list the advantages and disadvantages of each model. The detailed information are show as the yellow highlighted part in Section 5.5 in our revised manuscript. Table 3 shows that all models have some advantages, but the disadvantages of some models are not introduced, mainly because most of these models are applied to specific scenarios and solve specific application problems. Since the researches in the field of multi-modality event detection faces different scenarios and the problems to be solved are also different, these models are not universal. Most of the researches are carried out on the self-collected datasets, so it is difficult to evaluate which model is the best or worst. Only a small part of studies were evaluated under the same dataset, and their performance were compared in the evaluation part of Section 7. In addition, we added some main issues and challenges for future research, shown as the yellow highlighted part in Section 9.
